# Applications of Carbon Nanotubes in Bone Regenerative Medicine

**DOI:** 10.3390/nano10040659

**Published:** 2020-04-02

**Authors:** Manabu Tanaka, Kaoru Aoki, Hisao Haniu, Takayuki Kamanaka, Takashi Takizawa, Atsushi Sobajima, Kazushige Yoshida, Masanori Okamoto, Hiroyuki Kato, Naoto Saito

**Affiliations:** 1Department of Orthopaedic Surgery, Okaya City Hospital, 4-11-33 Honcho, Okaya, Nagano 394-8512, Japan; 2Physical Therapy Division, School of Health Sciences, Shinshu University, 3-1-1 Asahi, Matsumoto, Nagano 390-8621, Japan; kin29men@shinshu-u.ac.jp; 3Institute for Biomedical Sciences, Interdisciplinary Cluster for Cutting Edge Research, Shinshu University, 3-1-1 Asahi, Matsumoto, Nagano 390-8621, Japan; hhaniu@shinshu-u.ac.jp (H.H.); saitoko@shinshu-u.ac.jp (N.S.); 4Department of Biomedical Engineering, Graduate School of Medicine, Science and Technology, Shinshu University, 3-1-1 Asahi, Matsumoto, Nagano 390-8621, Japan; 5Department of Orthopaedic Surgery, Shinshu University School of Medicine, 3-1-1 Asahi, Matsumoto, Nagano 390-8621, Japan; kam17@hotmail.co.jp (T.K.); takashitak@shinshu-u.ac.jp (T.T.); ky2432@cumc.columbia.edu (K.Y.); ryouyuma@shinshu-u.ac.jp (M.O.); hirokato@shinshu-u.ac.jp (H.K.); 6Department of Orthopaedic Surgery, Marunouchi Hospital, 1-7-45 Nagisa, Matsumoto, Nagano 390-8601, Japan; soba@shinshu-u.ac.jp

**Keywords:** bone defects, scaffolds, carbon nanotubes, bone regeneration

## Abstract

Scaffolds are essential for bone regeneration due to their ability to maintain a sustained release of growth factors and to provide a place where cells that form new bone can enter and proliferate. In recent years, scaffolds made of various materials have been developed and evaluated. Functionally effective scaffolds require excellent cell affinity, chemical properties, mechanical properties, and safety. Carbon nanotubes (CNTs) are fibrous nanoparticles with a nano-size diameter and have excellent strength and chemical stability. In the industrial field, they are used as fillers to improve the performance of materials. Because of their excellent physicochemical properties, CNTs are studied for their promising clinical applications as biomaterials. In this review article, we focused on the results of our research on CNT scaffolds for bone regeneration, introduced the promising properties of scaffolds for bone regeneration, and described the potential of CNT scaffolds.

## 1. The Bone Healing Process

### 1.1. Current Concepts of Bone Regeneration Medicine

Bones are tissues that support the weight of an organism and serve as fulcrums of movement as they provide points of attachment for muscles. Although bone tissues are naturally capable of self-healing, the process is difficult with complex fractures and huge bone defects, potentially resulting in non-unions. Fracture healing is generally classified into direct healing without callus formation that occurs under precise repair and fixation, and indirect healing with callus formation that enables the bridging of bony fragments. Indirect fracture healing is the most common form of fracture healing [1]. Immediately after trauma, the first few days of healing represent an inflammatory stage. During this stage, bleeding occurs as a result of blood vessel disruption in the bone marrow, bone cortex, and periosteum, which leads to hematoma formation. Aggregated platelets initiate the secretion of tumor necrosis factor (TNF)-α and interleukin (IL)-1,6 [2], and inflammatory cells such as neutrophils and macrophages begin to migrate [3]. At the same time, mesenchymal stem cells (MSCs) migrate from the bone marrow and surrounding tissues to differentiate into osteoblasts and chondrocytes [4]. Cytokines such as bone morphogenetic protein (BMP)-2 and BMP-7 are believed to be involved in the differentiation of MSCs [5]. Local oxygen deprivation due to vascular damage stimulates the expression of hypoxia inducible factor (HIF), which induces vascular endothelial growth factor (VEGF) [6]. The reparative stage begins during the next few weeks. The ends of the bone are joined by a soft callus that is mainly composed of fibrous bone, and calcium is deposited on the osteoid tissue. Further ossification results in the formation of hard callus [7]. Although the abundant formation of callus during this stage may make the bones appear to be thick, its structural strength is weak [8]. The final stages of fracture healing occur over the following months to years and is known as the remodeling stage. During this stage, angiopoietin and VEGF-mediated pathways promote the angiogenesis of woven bone [9], which is subsequently replaced with more rigid lamellar bone. Bones are continuously remodeled via repeated cycles of bone resorption by osteoclasts and bone formation by osteoblasts, thereby restoring the original bone structure as a result of decreased callus volume and increased mechanical strength due to calcification [8].

### 1.2. Bone Regenerative Medicine

Regenerative medicine based on tissue engineering has been widely researched for non-unions that are difficult to treat with existing treatments and large bone defects for which an effective treatment option has not been established [10,11]. The combination of (1) cells, (2) growth factors, and (3) scaffolds are key elements in regenerative medicine, and the development of scaffolds must consider the relationship between cells and growth factors [12].

Three major types of stem cells are currently available in regenerative medicine: Somatic stem cells that naturally exist in human bodies, embryonic stem (ES) cells that are derived from culturing embryos (fertilized eggs), and induced pluripotent stem (iPS) cells that are artificially produced. Among these three types of stem cells, somatic stem cells have advanced the most in terms of their application to medical treatment. Because somatic stem cells are derived from the human body, they possess features that are ideal for therapeutic applications. There are several types of somatic stem cells, such as neural stem cells, mesenchymal stem cells, and hematopoietic stem cells. Mesenchymal stem cells have the ability to differentiate into various tissues and organs such as bone, cartilage, and fat cells [13]. Scaffolds in regenerative medicine must be able to provide an environment for these cells to proliferate efficiently.

Growth factors are chemicals that encourage cells to differentiate and proliferate. Both BMP-2 and -7 play important roles in the differentiation and proliferation of osteocytes. Although growth factors such as the fibroblast growth factor (FGF), transforming growth factor (TGF)-β, insulin-like growth factor 1 (IGF)-1, and platelet-derived growth factor (PDGF) do not exhibit an ectopic osteoinductive capacity, they do have osteogenic properties [14,15]. Immediately after a fracture, these molecules are secreted around defected tissues that require regeneration, and bone marrow-derived mesenchymal stem cells are differentiated and proliferated. Bone regeneration, however, cannot be obtained if the defect is too large or growth factors are deficient; thus, it is necessary to use a combination of a scaffold and a growth factor. Techniques for retaining endogenous growth factors and efficiently delivering artificially synthesized growth factors in cells of the body are known as drug delivery systems (DDS) [16]. Regardless of whether the technique is performed in vivo or ex vivo, a DDS enables the controlled delivery of unstable growth factors with no specific site of action when combined with biomaterials, thereby obtaining its maximum biological activity [17]. Growth factors for regenerative therapy are proteins that enhance the ability of cells to proliferate and differentiate, and scaffolds are studied as potential carriers for delivery of these proteins to cells.

Various structures have been studied for scaffolds in bone regenerative medicine. Materials can be broadly divided into biodegradable and nonbiodegradable materials. Collagen [18], poly-l-lactic acid (PLLA) [19], resorbable bioactive glass (BG) [20], and bioresorbable ceramics (BCs) including β-tricalcium phosphate (TCP) [21] are used as biodegradable materials, while metal (e.g., titanium) [22,23], carbon [24], and polyetheretherketone (PEEK) [25] are used as nonbiodegradable materials. Advantages of biodegradable scaffolds lie in the osseous replacement that occurs at the same time as bone regeneration, providing a more biomimetic regeneration with less risk of infections and foreign body reactions; however, when used for filling large bone defects, residual bone defects may remain. Nonbiodegradable scaffolds do not replace bone and can serve as a spacer or bone substitute to fill bone defects and transmit load (Figure 1).

### 1.3. Materials Used in Bone Regenerative Medicine

The current gold standard for treating large bone defects following tumor resection or trauma is autologous bone grafting [26]. Autologous bone grafting is considered an ideal material as it fulfills all three key elements of regenerative medicine [27]. However, some notable shortcomings include the limited amount of available material and pain at the harvest site [28]. Although allogeneic bone is relatively abundant and may be used for large defects, it can also potentially activate an immune response and may also pose some difficulty in engrafting [29,30]. Artificial bone made of hydroxyapatite [31] is widely used clinically due to its biocompatibility and ease of industrial production. Moreover, allogeneic bone grafting may only function as scaffolds due to cell death and altered growth factors. Artificial bone made of hydroxyapatite, PLLA, collagen, or their composite material also function as scaffolds [32]; however, since there are only several types of materials that can be used for artificial bones, it is not possible to completely reproduce the host tissue they intend to help regenerate.

The development of scaffolds is vital for regenerative medicine, and there has been a growing body of research on the use of carbon nanotubes (CNTs) as scaffolds [33]. In the field of bone regenerative medicine, an in vitro study in 2002 showed that the CNT/polylactic acid composite promoted an increase in osteoblast proliferation [34,35]. Subsequent studies have reported that CNT/polycarbonate urethane and CNT/poly (lactic-co-glycolic acid) (PLGA) composites enhance osteoblast adhesion [36,37]. In 2006, the use of CNTs alone were shown to promote bone and osteoblast proliferation [38], and several in vitro studies have since been reported to show its specialized functions on bone-related cells [38,39,40].

Scaffolds for bone regenerative medicine require a material in which cells proliferate, promote differentiation by growth factors, and maintain mechanical strength with equal or superior results compared to autologous bone grafts. In this paper, we will discuss the potential of CNTs as a scaffold in bone regenerative medicine by comparing their cell affinity, chemical properties, mechanical properties, and safety with currently used materials.

## 2. Recent Biomaterial Studies in Bone Regeneration Medicine

### 2.1. Cell Affinity of Scaffolds

In discussing the surface properties of biomaterials, it is necessary to consider their cell affinity. Nanostructured materials with a suitable surface for cell growth can promote cell growth more effectively than conventional materials [41,42]. However, the appropriate size of its particles, fibers, and pores varies from cell to cell [43,44,45]. Another important feature of the scaffold is its three-dimensional structure. In order to repair bone, it is necessary for cells to efficiently enter the scaffold, produce bone matrices, fill the interior, and crosslink to the bony cortex on the opposite side [46]. Therefore, pores inside the scaffold are necessary, and pore interconnectivity is desirable. Pore sizes between 300–500 µm were demonstrated to promote bone regeneration [47]. Scaffolds are made by various methods, including solvent casting, gas forming, and freeze drying; furthermore, in addition to these conventional methods, novel techniques such as three-dimensional molding with ink jet bioprinting technology are also being explored [48]. Angiogenesis is also an important factor for consideration. If there is no vascularization in the grafted bone, the bone will not survive. Although this problem could be addressed with vascularized bone grafting, the procedure can be complex and prone to surgeon error. In order to promote angiogenesis in a nonvascularized material, new blood vessels must be regenerated within the material. Interconnected porous hydroxyapatite has demonstrated excellent angiogenic properties. It has a porosity of 75%. The average pore size diameter is 150 µm and the average interpore connections’ diameter is 40 µm [49]. Recent studies have reported excellent results in new scaffolds with unidirectional interconnection. It has a porosity of 75–84% and a porous network with a diameter of 100–350 µm [46,50].

### 2.2. Chemical Properties of Scaffolds

Scaffolds require the in vivo ability to adsorb and retain growth factors that are secreted by their own cells [51] and to facilitate the sustained release of artificial growth factors [52]. Promising materials for bone regenerative medicine include scaffolds made of biodegradable materials such as PLGA [53] and hydrogel [54], in addition to porous hydroxyapatite scaffolds with coated surface modifications [55], all of which have been reported to provide a good sustained release of growth factors such as BMP-2. Moreover, there are current studies on stimulating local BMP-2 production via cell-seeded scaffolds with BMP-2 genes to induce bone formation [56].

### 2.3. Mechanical Properties of Scaffolds

Optimal intercellular contact in 3D cell culture, mechanical stimulation, and sufficient culture media and growth factors must be achieved in order to create long-term functional tissue in vivo after transplantation. Various bioreactors have been developed and used as devices for satisfying the above conditions and achieving efficient 3D cell culture [57]. As stated in Wolff’s law [58], functional remodeling reorients the trabeculae so as to align with new principal stress trajectories when the environmental loads on the bone are changed by trauma, pathology, or change in life pattern. Therefore, in order for bones to regenerate, mechanical conditions must approximate the conditions under which the original bones were placed. The compressive strength of the compact human bone is about 133 MPa loaded normal to the bone axis and about 170–193 MPa loaded parallel to the bone axis [59]. Scaffolds for bone regeneration are required to function as a place to reproduce these mechanical conditions. Natural polymers such as collagen are soft but offer high biocompatibility, and synthetic polymers such as polylactide (PLA) are hard and elastic but are less biocompatible. Currently, natural polymers are used as scaffolds for biological tissues that do not require high mechanical strength, while hydroxyapatite, β-tricalcium phosphate (β-TCP), and other synthetic polymers or natural polymer composites are used as scaffolds for tissues that require structural integrity such as bone [60,61,62].

### 2.4. Safety of Scaffolds

Safety is one of the key requirements for scaffolds in bone regenerative medicine. For a scaffold to be safe, the scaffold must either (1) degrade in the body or (2) remain without harming the body. Most materials used today, including metals and PEEK, rarely cause harm when placed inside the body. However, these materials may cause unintended infection or foreign body reactions such as metal allergies. If safety is of primary concern, a biodegradable material would be desirable. However, there is a trade-off between resorption rate and strength, and it is necessary to develop a material that retains enough strength for the time required for bone regeneration [63].

## 3. CNTs in Bone Regeneration Medicine

### 3.1. Cell Affinity of CNTs

Osteoblastic ROS 17/2.8 cells [38] and osteoblast-like cells (SaOS2) [64] have demonstrated good cell proliferation on CNTs. It is also reported that the human osteoblast cells and adipose-derived MSCs showed good cell proliferation with small restriction on CNT-based scaffolds [65,66]. There is also a report that carboxylic acid (COOH) functionalized multi-walled carbon nanotubes (MWCNT) showed good cell proliferation and differentiation [67], demonstrating that CNT-based materials have surface properties with good cellular affinity. Good cell proliferation was also shown in studies that evaluated these materials as 3D scaffolds for bone regeneration [65]. CNTs have been reported for their toxicity in certain types of cells [68]. However, intercellular accumulation of CNTs must exceed a certain level for the material to cause cytotoxicity [69,70,71].

### 3.2. Chemical Properties of CNTs

Hydroxyapatite (HA) crystals precipitate around osteoblast-secreted type I collagen to regenerate bone [72]. MWCNTs can precipitate HA crystals [73], and this property is believed to enhance bone affinity by efficiently depositing HA that serves as a bone matrix at the site of bone regeneration. In addition, we have previously reported that MWCNT blocks showed excellent protein adsorption [51,74] (Figure 2). In addition, many studies have reported that MWCNTs showed good adsorption of proteins such as BSA (bovine serum albumin) and immunoglobulin G (IgG) [75], suggesting that it is an excellent scaffold for regenerative medicine in terms of its ability to maintain and provide a sustained release of growth factors. Carbon nanotubes are also reported to adsorb proteins electrostatically or by hydrophobic interaction [75], and there are reports that CNTs adsorbing proteins are phagocytosed by macrophages and are easily degraded [76]. Nanoparticles such as CNTs that are exposed to the internal environment of the body were found to be rapidly coated by proteins, and it has become clear that the protein adsorption layer is an important factor in determining the biokinetics of nanoparticles [77]. Protein surfaces are heterogeneous in terms of their morphological and physicochemical properties, and various factors such as hydrogen bonding, electrostatic interaction, hydrophobic interaction, and van der Waals force are involved in their adsorption to other molecules. These interactions are traced back to the 20 amino acids that function as building blocks of proteins. Aromatic amino acids (Trp, Tyr, Phe) have been shown to interact more strongly with CNTs than other hydrophobic amino acids [78]. The aromatic amino acid side chains interact with CNTs and are oriented parallel to the CNT surface. The strong adsorption on the CNT surface is believed to be due to π-π interactions. Among charged amino acids, Arg has been shown to have a higher dispersion effect than Lys and Glu [79]. Although Arg is highly hydrophilic, it binds to highly hydrophobic CNTs, a feature not found in other amino acids. Arg is said to promote osteoblast differentiation [80], and a chemical modification using Arg may promote bone formation. CNTs also bind to various molecules such as antibodies, radioisotopes, and anticancer drugs, and have been studied as a promising carrier for DDS [81].

### 3.3. Mechanical Properties of CNTs

Due to their microscopic properties, it is difficult to produce flexible 3D structures out of nanocarbon materials. Cortical bone is an elastic and plastic substance with tensile strength of 0.051–0.133 gigapascals (GPa) and a Young’s modulus of 12–18 GPa. In contrast, MWCNTs have a tensile strength of over 63 GPa and Young’s modulus of 200–1950 GPa [82], but its macroscopic strength as a material can change according to the molding technique that is implemented. In our previous study, we demonstrated that blocks made of MWCNTs alone are too soft with porous interiors [51] and too hard without pores [74]. On the other hand, bones are composite materials made predominantly of collagen and hydroxyapatite. It is difficult to reproduce the characteristic strength of bones with one material alone, and it is considered desirable to combine CNTs with multiple materials as a bone replacement material that reproduces their strength and elasticity [83]. To date, various studies have reported good results on scaffolds made of composite materials, including PLA and hydroxyapatite [84], polyglycolic acid (PGA) and βTCP [85], and sponge-like structures composed of collagen inside the porous bodies of PLA and PLGA [84,86]. Nanocarbon- and polymer-based composites include 3D collagen scaffolds coated with MWCNTs [87] and scaffolds made from a composite of CNTs, collagen, and mineral trioxide aggregate (MTA) [88] (Figure 3). When rat primary osteoblasts were cultured onto a MWCNT-coated collagen scaffold, the calcium content in the scaffold and osteopontin content, which is a bone morphogenetic factor, were higher than those in the collagen scaffold. Moreover, in an experiment in which MWCNT-coated collagen scaffolds and collagen scaffolds were implanted in femoral bone tunnels of rats, the MWCNT-coated collagen scaffold was replaced by more bone tissue, and MWCNTs were incorporated into the bone matrix. By incorporating CNTs into the bone matrix, they can function as fillers and potentially improve the mechanical properties of bone. Furthermore, nanoshaped materials or surfaces can inhibit bacteria adhesion due to cytoskeleton stress [89]. Because bacterial infection is a common cause of implant failure in bone, CNTs are also expected to have antibacterial properties [90]. That is a strong point to support the use of CNTs for biomaterials aimed at bone repair.

### 3.4. Safety of CNTs

CNTs are considered physicochemically stable and show minimal degradation [91]. Therefore, further biosafety testing, such as long-term pharmacokinetics tests of scaffolds, is needed. In our own research, subcutaneous injections of MWCNTs into carcinogenic rasH2 mice were found to be less carcinogenic than carbon black, a negative control [92]; intra-articular injections in knees of rats caused no long-term inflammation [93]; and intravenous injections showed no organ accumulation or carcinogenicity [94] (Figure 4). It has also been shown that CNTs can either unintentionally or intentionally create molecular defects during fabrication, which can be attackable sites for microbial and enzymatic degradation [95]. Enzymes and bacteria that degrade carbon nanotubes have also been reported [96,97]. 

However, carbon nanomaterials’ biodistribution still remains unclear due to their size, shape, and physical-chemical properties. When aspirated, MWCNTs cause fibrotic response in the alveolar tissues of the lungs [66]. MWCNTs also induce inflammation and granulomas when injected into the abdominal cavity [98]. When using CNTs as scaffolds, it is necessary to devise ways to prevent the fibers from being incorporated into cells, and further research is needed on their long-term biosafety and degradability.

## 4. Our Research on CNT Scaffolds

Usui et al. first revealed in 2008 that CNTs promote bone tissue formation in vivo [99]. Narita et al. later reported that CNTs suppressed the receptor activator of nuclear factor-κB (NF-κB) ligand (RANKL) expression and suppressed osteoclast differentiation [100]. Shimizu et al. revealed that CNTs can activate osteoblasts and promote the calcification of bones [101]. These studies suggested that CNTs could be used to create a scaffold that promotes bone formation; however, there were no studies to our knowledge that described 3D scaffolds composed exclusively of CNTs for bone regenerative medicine. We created a 3D block structure composed of MWCNTs with maximized mechanical strength (MWCNT blocks) [102,103] and evaluated their efficacy as scaffold material for bone regeneration [74]. The densities of the MWCNT blocks were equivalent to bones [104], and the Vickers yield strength was comparable to cortical bone [103]. The Young’s modulus of the MWCNT blocks was about one-half to two-thirds that of cortical bone, and their bending strength (29.0–47.4 megapascals (MPa) ) was about one-quarter to one-third that of cortical bone [105]. MWCNT blocks containing 5 μg of recombinant human bone morphogenetic protein-2 (rhBMP-2) was implanted in the back muscle of mouse and evaluated by microcomputed tomography (μCT) at 3 weeks after implantation. Ectopic bone formation was confirmed on the positive control PET-reinforced collagen scaffolds containing BMP-2, and similar ectopic bone formation was observed on MWCNT blocks with rhBMP-2 (Figure 5a). In cell adhesion tests on MWCNT blocks, positive-control polyethylene terephthalate (PET) reinforced collagen scaffolds and coverslips. In the cell adhesion test using three types of cell cultures on the MWCNT blocks and the positive-control coverslip, pronounced cell adhesion on the MWCNT blocks was observed on Day 1 and Day 3 in V79 cells (fibroblasts) and RAW267.4 cells (macrophages). In addition, we assessed the performance of the CNT porous block (CNTp) [106], which was composed of MWCNTs that were molded into a porous 3D scaffold for bone regeneration to maintain biocompatibility, osteoconductive ability, and bone morphogenetic proteins [51]. CNTp showed excelling cell adhesion and proliferation. In an in vivo experiment, CNTp implanted in mouse calvarial bone defects showed bone regeneration and repaired bone defects (Figure 5b). Furthermore, the addition of rhBMP-2 resulted in a greater degree of bone formation. This indicated that CNTp exhibited excellent bone affinity as a scaffold and was also excellent as an agent to provide a sustained release for rhBMP-2. Since the MWCNT block used in this test was cauterized and solidified, we believe that there was almost no release of MWCNTs into the aqueous solution with virtually no possibility of being incorporated into cells. In our in vitro studies, MWCNT blocks were not toxic to the three representative cell types [74]. 

## 5. Conclusions

CNTs are believed to promote the proliferation and differentiation of osteogenic cells due to their protein adsorption and chemical modification. In addition, the material may be a promising carrier for DDS as it can bind to artificial bone morphogenetic proteins. Considering their mechanical properties and nanostructure, CNTs show considerable promise as a scaffold for bone regeneration with high bone affinity and safety for use in bone tissues (Figure 6). In order to use the material as a scaffold, it is necessary to assess and establish their long-term safety, produce composite materials, or devise a strategy to reproduce biomimetic properties with a three-dimensional structure.

## Figures and Tables

**Figure 1 nanomaterials-10-00659-f001:**
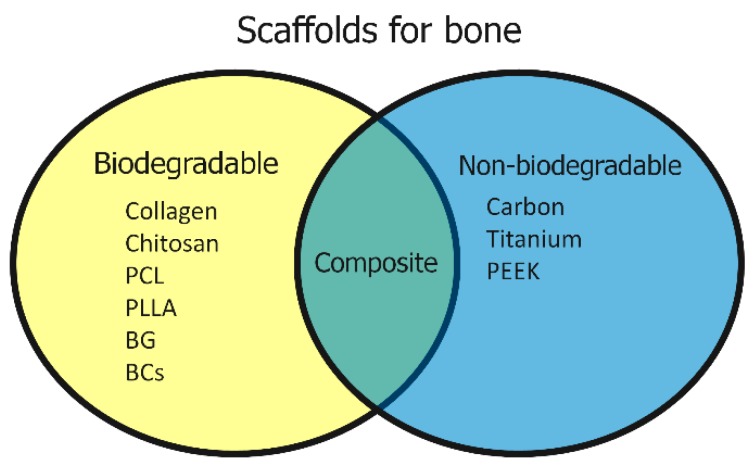
Scaffolds for bone regeneration include biodegradable and nonbiodegradable scaffolds, as well as composite scaffolds comprised of a combination of each material. PCL (polycaprolactone), BCs (bioresorbable ceramics), BG (bioactive glass), PLLA (poly-l-lactic) acid, PEEK (polyetheretherketone).

**Figure 2 nanomaterials-10-00659-f002:**
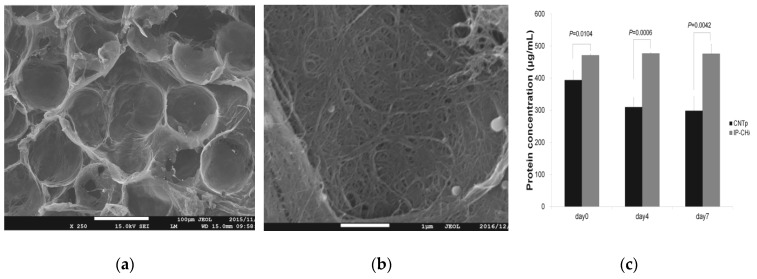
(**a**) Scanning electron microscope (SEM) image of a carbon nanotube porous block (CNTp). Scale bar, 100 μm. (**b**) SEM image of CNTp. Scale bar, 10 μm. (**c**) When a scaffold composed of interconnected porous hydroxyapatite ceramics (IP-CHAs) was immersed in a bovine serum albumin (BSA) solution, more protein was adsorbed to CNTp and the protein concentration in the solution was reduced. Image is modified from a study by Tanaka et al. [51].

**Figure 3 nanomaterials-10-00659-f003:**
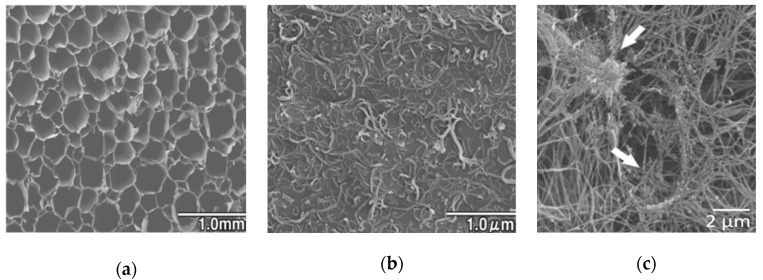
(**a**) SEM image of multi-walled carbon nanotubes (MWCNT)-coated collagen scaffold, (**b**) SEM image of the internal pores of MWCNT-coated sponge. Fibers can be observed on the surface of MWCNTs. Reproduced with permission from [87] Elsevier, 2011. (**c**) SEM image of collagen/MWCNT composite scaffold. MWCNT aggregates were confirmed within the collagen fiber (white arrows). Reproduced with permission from [88] Elsevier, 2016.

**Figure 4 nanomaterials-10-00659-f004:**
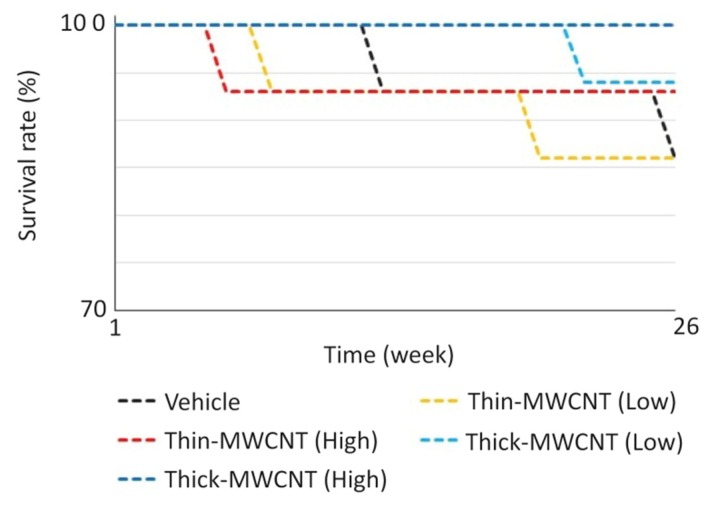
Survival curve of rasH2 mouse with intravenous administration of MWCNTs. The experiment was performed with n = 15 in each group. Even in the group without MWCNT administration, no significant difference was found between the vehicle group and the mortality rate. Image is modified from a study by Sobajima et al. [94].

**Figure 5 nanomaterials-10-00659-f005:**
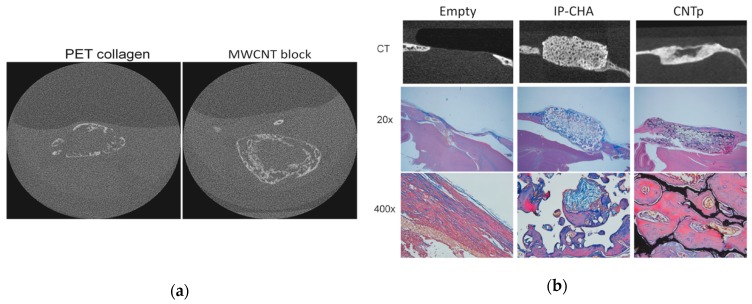
(**a**) The microcomputed tomography (μCT) image of ectopic bone formed at three weeks after subcutaneous implantation of scaffold with recombinant human bone morphogenetic protein-2 (rhBMP-2) in a mouse model. Ectopic bone formation comparable to that of the MWCNT block was found in the polyethylene terephthalate (PET)-reinforced collagen scaffold. Image is modified from a study by Tanaka et al. [74]. (**b**) In a mouse calvarial defect model, critical size bone defects were repaired with new bone in both the IP-CHA and CNTp groups. Image is modified from a study by Tanaka et al. [51].

**Figure 6 nanomaterials-10-00659-f006:**
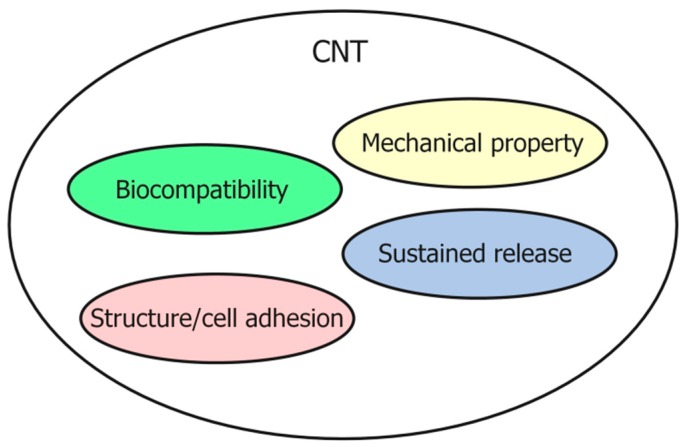
Characteristics of CNT scaffolds. The material properties of CNTs are suitable for bone regeneration scaffolds due to their biocompatibility, structure, mechanical properties, and ability to provide sustained release.

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
