# Peer review of "Applications of Carbon Nanotubes in Bone Regenerative Medicine"

_nanomaterials, 2020, doi:10.3390/nano10040659_

Round 1
Reviewer 1 Report
The review presented by Tanaka et al is an interesting update about the use of carbon nanotubes for bone repair.
The manuscript is well written and the topic is surely of interest for the readers of this Journal; accordingly, I have some minor comments prior to suggest this Review for publications.
Here my comments:
- Figure 1: I suggest to add (and maybe briefly describe in the text) some other common materials for bone repair such as bioactive glass and ceramics.
- Chapter 2.1: I suggest to better specify the shapes of the cited nanomaterials .
- Chapter 2.3: I suggest to detail the specific mechanical parameters necessary to support bone compressive stress during healing that must be considered for biolmaterials design.
- Chapter 3.1: the Authors mentioned interesting works involving mature osteoblasts. I suggest to include some other papers dealing with non immortalized cells (such as foetal pre-osteoblasts hFOB) or stem cells that are more sensitive to biomaterials properties.
- Chapter 3.4: this part is in my opinion a bit understimated. Nanomaterials (in general) biodistribution still represents a challenge due to their size, shape and physical-chemical properties; the Authors should better describe also the negative outcomes that are related to nanomaterials injection.
- Nanotubes antibacterial properties: due to antibiotic resistance, bacterial infections represent a common reason of implant failure in bone despite the biocompatibility of the materials. Nanoshaped materials or surfaces hold intrinsic ability to inhibit bacteria adhesion due to cytoskeleton stress as showed for example in doi:10.3389/fbioe.2019.00103. This is a strong point to support the use of nanotubes for biomaterials aimed at bone repair and I think that it is useful to mention it in the text of this Review.
Author Response
Response to Reviewer 1 Comments
Point 1: Figure 1: I suggest to add (and maybe briefly describe in the text) some other common materials for bone repair such as bioactive glass and ceramics.
Response 1: Thank you for your advice. We added examples of bioactive glass and ceramics. Cause we consider beta TCP as a kind of bioresorbable ceramic(BC)s, we added BC instead of beta TCP (page 3, line 101-102, Figure 1).
Point 2: I suggest to better specify the shapes of the cited nanomaterials.
Response 2: We added explanations about the shape of them (page 4, line 157-160).
Point 3: I suggest to detail the specific mechanical parameters necessary to support bone compressive stress during healing that must be considered for biolmaterials design.
Response 3: We added an information about that (page 5, line 178-180).
Point 4: the Authors mentioned interesting works involving mature osteoblasts. I suggest to include some other papers dealing with non immortalized cells (such as foetal pre-osteoblasts hFOB) or stem cells that are more sensitive to biomaterials properties.
Response 4: We added informations about human osteoblast cells and adipose derived MSCs (page 5, line 201-202).
Point 5: this part is in my opinion a bit underestimated. Nanomaterials (in general) biodistribution still represents a challenge due to their size, shape and physical-chemical properties; the Authors should better describe also the negative outcomes that are related to nanomaterials injection.
Response 5: We added our comments about them (page 7, line 290-293).
Point 6: Nanotubes antibacterial properties: due to antibiotic resistance, bacterial infections represent a common reason of implant failure in bone despite the biocompatibility of the materials. Nanoshaped materials or surfaces hold intrinsic ability to inhibit bacteria adhesion due to cytoskeleton stress as showed for example in doi:10.3389/fbioe.2019.00103. This is a strong point to support the use of nanotubes for biomaterials aimed at bone repair and I think that it is useful to mention it in the text of this Review.
Response 6: We added our comments and recommended citations (page 6, line 264-267).
Reviewer 2 Report
The review is correctly carried out, with an adequate and updated list of the properties of the CNTs that make them useful for their application to bone regeneration, with the reference to the most recent studies and the novel techniques that have been implemented in This field, exposed in a sufficiently clear way, and with a wording that arouses the interest of the reader, ending with the exposition of their own work, which demonstrates some graphically proven success.
Therefore, I consider that the article is suitable for publication in its current form.
Author Response
 We would like to express our gratitude for reviewing our manuscript and providing us with valuable advice to improve our paper. We have revised our manuscript according to the reviewers' comments. We believe that these revisions have greatly enriched our manuscript in its content and presentation, and we hope that you will now deem it suitable for publication in nanomaterials.